# EVOLVING NEURAL UPDATE RULES FOR SEQUENCE LEARNING

## ABSTRACT

We consider the problem of searching, end to end, for effective weight and activation update rules governing online learning of a recurrent network on problems of character sequence memorization and prediction. We experiment with a number of functional forms and find that the performance depends on them significantly. We find update rules that allow us to scale to a much larger number of recurrent units and much longer sequence lengths than has been achieved with this approach previously. We also find that natural evolution strategies significantly outperforms meta-gradients on this problem, aligning with previous studies suggesting that such evolutionary strategies are more robust than gradient back-propagation over sequences with thousands(s) of steps.

## 1 INTRODUCTION

The field of neural sequence processing has become dominated by neural networks trained by back-propagation, with the best models being transformers currently, and LSTM recurrent networks in the not so recent past. These networks can be used for a range of problems such as label prediction, modelling and reinforcement learning. The basic computational nature of these networks can be characterized as follows: One can design an arbitrary (differentiable) forward computation that updates part of the network (activations) and then one executes a specific algorithm to do the updates of another set of parameters (weights - using back-propagation and variants of stochastic gradient descent). This has an advantage of freedom to flexibly design a forward computation. However, the hand-coded update algorithm is fixed, and may not be the most effective means of achieving a given training objective. There has been interest, as we review below, in directly finding the full computation (both activations and weights updates). In this paper, we aim to push this approach further by finding update rules that work better and that scale, by comparing different functional forms and methods for optimizing the search for these rules.

We pursue what we call end-to-end search for update rules. A good way to explain this is to consider the analogous process in nature. Roughly speaking the brain of an organism is produced from the information of the parents, "run" for the lifetime of the organism, and selected based on how it, as well its descendants perform (for example keep making descendants). Over time, this process developed neural networks that can learn, adapt quickly and that possess whichever other abilities were needed for the success of the organisms. Similarly we consider a space of neural networks and search for updates to their parameters, the activations and the weights in our case, that work well on problems we are interested in. This can allow networks to discover whatever updates are needed to solve these problems.

We consider parameterization of neural networks that is somewhat similar to classic artificial recurrent neural networks, with a state consisting of activations $h_t$ and weights $w_t$ at time $t$. There are, however, two primary differences. The first is that the entire computation (including learning or whatever computation the network does) operates online, that is, the network receives input $x_t$ and updates according some function

$$h_t, w_t, p_t = f_\theta(x_t, h_{t-1}, w_{t-1}) \tag{1}$$

where $f_\theta$ is the update function parameterized by hyper-parameters $\theta$ that are fixed for the lifetime of the network. The $p_t$ is the output - in our case the probability over the next character in the sequence (we will discuss the problems we solve later). The second difference is that the activation $h_{t,i}$ of a

given neuron $i$ as well as the weight $w_{t,ij}$ between neurons $j$ and $i$ are (potentially) small vectors, as in (Bertens & Lee, 2019; Gregor, 2020).

Biological neurons were an inspiration for artificial neural networks, and so is the case here. The former are complex objects, and representing the cell body state as well as the synapse state by vectors (rather then scalars) should allow for a closer representation of the neuron's computation. Biological neurons furthermore provide a proof of principle that online neural computation is capable of powerful learning.

There are several additional motivations for pursuing this approach. Learning general update rules end to end might allow networks to use weights in a more interesting or powerful fashion than that prescribed by back-propagation. Finding online rules might allow for easier hardware implementation, especially in neural hardware architectures (Modha, 2017; Davies et al., 2018). Developing the ability to search for update rules might help us to discover how brains compute, for example if we can implement what we know about the computational structure and search for what we don't. Finally in artificial life, entities propagate into the future often not based on an objective but simply if they find means of making descendants (and if those descendants make more descendants and so on) (Ray, 1991; Aguilar et al., 2014; Soros & Stanley, 2014; Gregor & Besse, 2021). Being able to evolve learning algorithms without objective might provide a path to intelligent artificial life.

## 1.1 RELATION TO PREVIOUS WORKS.

*Evolving network weights*

There are many works that directly evolve network weights, for example (Wierstra et al., 2014; Salimans et al., 2017). In (Stanley & Miikkulainen, 2002) they parameterize the weights by compositional pattern producing networks (CPPN) and in (Risi & Stanley, 2010) they consider an adaptive version that updates the weights based on pre- and post-synaptic activations. While in one sense this is similar to our approach in that the weights are updated, it is also very different - with a very different parameterization (CPPN's and no hidden network units), and problems (evolving the system to solve a T-maze task), and without a focus on pure end-to-end learning as the parameters are also evolved for the task.

*End-to-end learning and meta-learning*

A range of related works explore end-to-end learning and meta-learning, but don't strictly operate in the local update rules setting. Furthermore, many of these works learn the full weights of the network instead of a smaller number of update rule hyper-parameters. In (Schmidhuber, 1992) they consider an end-to-end setting, parametrizing updates of a (fast) network, by another (slow) network, that outputs vectors used to compute Hebbian updates to parameters. In AutoML zero (Real et al., 2020) they evolve a sequence of operations such as multiplication and non-linearities that implement a machine learning algorithm (that includes learning).

A series of works such as (Ravi & Larochelle, 2016; Metz et al., 2019; Wichrowska et al., 2017; Bello et al., 2017; Li & Malik, 2017; Lv et al., 2017) focus on learning generalisations of gradient-descent optimisers that utilise back-propagated gradient information to update weights. Although these works optimise the parameters of the update rule over multiple update steps, the tasks themselves are typically feed-forward in nature, such as image classification. In some works (Ravi & Larochelle, 2016), they explicitly consider few-shot learning, which is related to online learning, although task relevant information is preserved across tasks in the full network weights.

In (Miconi, 2016; Miconi et al., 2018) they consider recurrent neural networks, but introduce per-weight state variables that are updated in a Hebbian fashion with parameters of the updates trained by back-propagation, along with a standard set of weights. This is capable of learning fast weight adaptations.

*Alternative network parametrizations*

In (Ha et al., 2016) they consider a recurrent network with weights parameterized by a smaller set of hyper-weights. This is trained by standard back-propagation over a short periods. Because the number of hyper-weights is still relatively large, the system is still relying on back-propagation learning to encode the structure of the current data-sequence, rather than a learning algorithm primarily.

*Learning local update rules end-to-end*

There are a number of works more closely related to our objective of end-to-end learning of local update rules (Bengio et al., 1992; Orchard & Wang, 2016; Gu et al., 2019; Munkhdalai et al., 2019; Bertens & Lee, 2019; Gregor, 2020; Kirsch & Schmidhuber, 2020). For example Bengio et al. (1992) parameterizes scalar weight updates as a linear combination of terms derived from local scalar activations and previous weights and searches for these parameters using genetic algorithms, stochastic gradient descent or simulated annealing applied to toy problems. In Orchard & Wang (2016) a population of agents is evolved for the task of foraging in a 2d world. However both the initial weights of the controller network as well as the synaptic update network are evolved, in principle putting good policies in initial weights (but showing that updates improve performance). Bertens & Lee (2019); Gregor (2020) introduce the idea of using vectors to represent both the activation and weight states, and updates using LSTM's on T-maze tasks in the former and MLP's on sequence memorization respectively in latter.

A key drawback in all these works is that the networks used are tiny, of usually less than ten neurons. In Gregor (2020), they attempt to use larger hidden layers, but find that the network does all its computation in the input layer. In Kirsch & Schmidhuber (2020) they consider more general network parameterizations, using LSTM-based local updates and scaling to a larger number of neurons. However they only consider problems such as MNIST classification that do not require an algorithm to learn from long-range time dependencies. That is, because the same classes often appear close to one another in a random sequence, there is a signal over just a few steps (longer range dependencies might be needed for good performance, however the reported one is low).

The main contribution of this paper is to find a parameterization of update rules and a method of search that scales to large number of neurons (we tested up to a thousand hidden units) and weights (million) and that can learn recurrent network training over long time spans (thousands).

## 2 TASKS

We train the networks on two tasks both defined on a sequence of characters of a text. We use the pg19 data-set (Rae et al., 2019) containing a large collections of books as our text source. In the first task we consider sequences of length $N$ taken at random points from the text. The $N$ is of the order of ten thousand in our experiments, but in principle it can be arbitrarily long or we should ideally use the entire data-set, this is just the scale we have managed to achieve so far. At each iteration, we initialize a neural network with random weights and zero activations, run it through this sequence online as in eq (1), and measure the total log-likelihood:

$$L = -\sum_{t=1}^{N} \log p(x_t|h_{t-1}) \tag{2}$$

We are looking for a set of hyper-parameters $\theta$ parameterising the update rule $f$ that minimizes the loss $L$. The resulting $\theta$ should encode activation and weight dynamics that learn a model of the sequence while running through it online. In the second task we play such randomly sampled sequence twice and measure the likelihood on the second part, testing network's ability to memorize.

## 3 OPTIMIZATION

There are two algorithms we compare for optimizing $\theta$. The first algorithm is (meta)-gradient, where we repeat the following process. We sample a mini-batch of randomly selected sequences from the text. We run the network forward and back-propagate, both through the whole batched sequence, and take stochastic gradient optimizer step - we use Adam (Kingma & Ba, 2014) in the experiments.

The second technique we employ is natural evolution strategies (NES) (Wierstra et al., 2014; Salimans et al., 2017). Instead of optimising only the parameters $\theta$, this method instead maintains a search distribution over the parameters. We use separable NES (SNES), in which this distribution is a diagonal normal distribution parameterised by the parameter means and variances. At each iteration, a population of update rules (population of $\theta$'s) is sampled from this distribution as well as a single sequence from the dataset. The networks are run forward to obtain fitnesses, which are then

used to estimate improved values of the parameter means and variances. In contrast to (Salimans et al., 2017), we find that updating the variances at each iteration is beneficial. Further details for our implementation are given in App. A.3.

## 4 MOTIVATION BEHIND THE TWO TASKS.

The standard way to compare models is on their likelihood on a withheld test set. However, this has its issues: For example, how much overlap (either direct or approximate) is there with the train set, how much out of distribution we want our models to generalize. A different view is to only have the data set, but compare models on how well they can compress it. For models, the compression essentially equals the likelihood of the data under the model, plus the cost to encode the model, or better, the code defining the model training and its decompression mechanism (Bellard, 2019; 2021). This is exemplified by the Hutter Prize (Hut) - a challenge to produce as small a file as possible, which when run, outputs a one billion character version of Wikipedia.

Here, let us consider a model that goes over the text (say over the letters), making next step predictions. If a model encounters a new piece of text, such as a word "computer", it needs to store some information about it. This is because the next time this word, or something related to it appears, perhaps many thousands of steps later, a good processing of the original word would allow the model to make good predictions. Standard likelihood training has a handcrafted means of storing the text: increase the likelihood of the current piece of text (replaying it many times, (Bellard, 2019)). This working relies on a generally reasonable *assumption* that statistics of data in the future will be similar to the current ones. In our approach we don't make such an assumption - the system evolves how it should process the current piece of information in order to do well in the future. This could in principle evolve both fast storage of the current piece of text, as well as a process similar to increasing the likelihood under a model over many time steps, in the same network. To do this, the search process needs to be able to find relationships that span thousands even possibly millions of steps. Thus, rather than hoping to find such signal by back-propagating over such periods, we can expect evolution to work better: For example, if local update rules are modified by chance to store information, this modification can have immediate effect over long time scales. To find strong relationships, we should ideally run over long text such as Hutter prize text. While we have not achieved such scale in yet, we show, for the first time, that the language model learning algorithm can be evolved end to end, even on a shorter pieces of text (order 10k).

In addition to prediction/compression, we would also like to have a task that has an adjustable difficulty level and clear performance measure; fast to experiment with for easier difficulties, and challenging for standard recurrent networks at harder difficulties. For this purpose we study sequence memorization: Presenting a sequence twice and measuring the likelihood on the second part. We define success at memorizing when the network likelihood is significantly smaller in magnitude (closer to zero) than pure prediction performance on just the text.

## 5 NETWORK DETAILS

In this section we describe in detail the operation of the neural networks that we use.

We use a three layer neural network in our experiments (though the formulation allows for arbitrary number of layers): The input layer $h^1$, the hidden layer $h^2$ and the output layer $h^3$. For any given network, each layer is of the shape $n^l \times n_a$ where $n_l$ is the number of neurons in layer $l$ and $n_a$ is the size of the activation state of a given neuron, usually six in our experiments (one in classic RNNs).

Layers are connected by weights. The weights going from layer $l$ to layer $k$ are denoted by $w^{kl}$ and form a tensor of shape $n^l \times n^k \times n_w$ where $n_w$ is the size of the state of a given weight. In classic RNN this is one in the forward pass, and it also ends up being one for the best performing networks, but we also experiment with more complex weight updates requiring a larger size.

The computation at a given time step is shown in the Algorithm 1. Here we explain the steps of this algorithm. The input $x$ is a one hot vector representing the input text character at the current time step, and is of size $n^1$ (equals the number of possible characters, 259 in our experiments). It is fed to the 0-th index of the layer 1 tensor: $h^1[:, 0]$ (so there are 259 neurons in the first layer, each with 6 dimensional (activations) state, and therefore the $h^1$ has shape $259 \times 6$).

---

**Algorithm 1** Network update at a given time step

---

**Require:** Input $x$
**Require:** Activations $h^1, \ldots, h^L$ of layers $l = 1, \ldots, L$ of shapes $n^l \times n_a$
   Normally: $L = 3$ with $h^1$ the input layer, $h^2$ the recurrent hidden layer, $h^3$ the output layer
**Require:** Weights $w^{lk}$ for each pair of layers that are connected
   Normally: $(1, 2), (2, 1), (2, 3), (3, 2), (2, 2)$
**Require:** Activations update $AU$
**Require:** Weights update $WU$
   $h^1[:, 0] = x$
   **for** $l = 1, \ldots, L$ **do**
      $a = [h^l]$
      **for** $k = 1, \ldots, L$ **do**
         **if** $l, k$ connected **then**
            $a.\text{append}(w^{lk} \cdot h^k[:, 0])$
            $a.\text{append}(w^{T,kl} \cdot h^k[:, 1])$
         **end if**
      **end for**
      $h^l = AU(a)$
      **for** $k = 1, \ldots, L$ **do**
         **if** $l, k$ connected **then**
            $W^{lk} = WU(w^{lk}, h^l, h^k)$
         **end if**
      **end for**
      $p = norm(relu(h^L[:, -1]) + 0.001)$
   **end for**

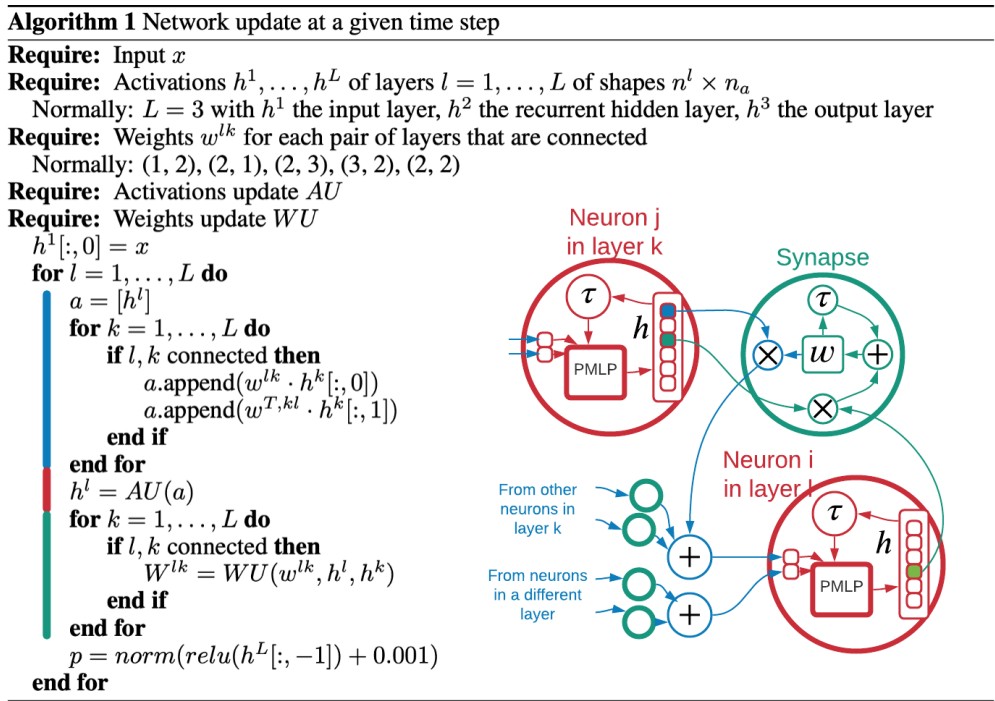

---

Figure 1: The network update at a given time step, Eq.1. The steps of the algorithm are explained in the text. The diagram shows the computation for the functional forms with the best performance, except $W^T$ is omitted for clarity. The $h$ is a six dimensional vector (the six red squares) representing the state of neuron (it's activation part). Blue: Communication between neural activations, consisting of matrix multiplications (multiplying $w$ and $h[0]$ in the figure and then adding with "+") that get appended to a vector $a$ (all the red squares within the neuron). Red: the neuron activations update (AU), containing hyper-weights $\theta$ inside of the product MLP (PMLP), Eq.3, that we evolve. The $\tau$ is the delay that returns previous value. Appending results from different layers (the two red squares on the left side of the neuron) and passing through the PMLP, allows for different layers to affect a given neuron differently, for example excitation from one layer and gating from another. Green: Hebbian like weight update (WU) taking product of component 2 of pre-synaptic and component 3 of a post-synaptic neuron. We can also consider PMLP and LSTM for synaptic updates (also containing evolved hyper-weights), however, we found that the evolved performance is worse as well as being computationally much more costly.

Next we iterate over the layers. At each layer, we first initialize a list of tensors $a = [h^l]$ to include the activations $h^l$ of the layer. We next check which layers are connected to the current layer. These are the layer above and the layer below (but could be other connectivity), if present, as well as a recurrent connection. We don't use recurrent connections in the input and output layers - only in the hidden layer(s). For a given connected pair of layers, we do two matrix multiplications (denoted by $\cdot$): $w^{lk}$ (that "goes" from layer $k$ to layer $l$) multiplies the 0-th component $h^k[:, 0]$ of tensor $h^k$ and the transpose of $w^{kl}$ (that "goes" from layer $l$ to layer $k$) multiplies the 1-st component $h^k[:, 1]$ of $h^k$. The results of the multiplications are added to the list $a$. After the $k$-th loop, the $a$ will contain multiplications from all the layers that the current layer is connected to, as well as the current value of its state $h^l$. Given this collection, we perform an activations update (AU), which produces the new activation state $h^l$. This is a point-wise update (without communication between neurons) and will be described in detail below.

Next, still at a given layer $l$, we go over all the layers $k$ and update the weights of the connections $w^{lk}$ that go from those layers to this one using a weight update $WU$ described in detail below. Again, this is an independent update for each weight.

Having updated the activations and weight at this time step we make a prediction of the next character. We take the last component of the last layer $h^3[:, -1]$ and compute the probability directly from it. We experimented with two versions. The first version is more standard - treating the neural vector as logits and computing the probability $p = softmax(h^3[:, -1])$. In the second version, we treat the neural vector as un-normalized probabilities and computing the probabilities by normalizing as shown in the algorithm.

Next, let us describe the activations update AU. The activations update can in principle be any function, usually parameterized by some parameters $\theta_a$. The function that we found to work well is a one hidden layer MLP with $\sigma \times \tanh$ non-linearity, which we denote by PMLP (product-MLP):

$$AU(a) = M_3 \cdot (\sigma(M_1 \cdot a) \tanh(M_2 \cdot a) + b_1) + b_3 \qquad (3)$$

where $M$'s are matrices and $b$'s are vectors and there is point-wise multiplication between the $\sigma$ and $\tanh$.

We also experimented with the LSTM non-linearity, where we split $h_t$ into the hidden state and cell state of the LSTM, treat the rest of $a$ as inputs and do a standard LSTM update.

Let us now describe the weights update WU. While we experimented with various forms of the update, we found that a simple Hebbian-like update with scalar weights for every connection worked the best:

$$WU(w^{lk}[i, j], h^l[i], h^k[j]) = w^{lk}[i, j] + h^l[i, 3]h^k[j, 2] \qquad (4)$$

where we have written the update for the weight going from neuron $j$ in layer $k$ to neuron $i$ in layer $l$. Importantly, note that we have used the index two component of the source neuron and index three component of the target neuron. That is, different components of the activation vector of a given neuron are used for different purposes: 0 for matrix multiplication, 1 for matrix multiplication by transpose, 2 in weight update for the source (pre-synaptic) neuron, 3 in the weight update for the target (post-synaptic) neuron (the specific numbers don't matter, only that they are different). This way the network can use different components as needed. Again, we also experimented with more complex forms, making $w$ a vector and using MLP and LSTM weight updates.

Here we explain the motivations behind various choices. We use a vector representation for neural activations as this provides a richer update while barely increasing computational cost, as it is of order $N$=number of neurons, not $N^2$=number of weights. We use a product MLP because a good update might require multiplications, but we don't know of which variables - PMLP can select which variables are needed. It can implement for example gating by one type of neuron (in one layer) and excitation by another type of neuron (in another layer), possibly matching closer the functionality of biological neurons. For the weights, we could have them between each pair of components of the activations vector, but that would significantly increase the cost - currently we have the same number of weights as a standard recurrent network - one per pair of neurons (actually somewhat more because connections go both ways). We use a Hebbian update rather than a more general parameterization by a neural network (even though we experiment with it as well) because again the computational cost is low - roughly equal to the classical $\tanh$ RNN with the same number of neurons (if we include back-propagation for the latter). We use $W^t$ to help with ability to imitate passing gradients. We use different components of the neuron activation for multiplication by $W$, for multiplication by $W^t$ as well as for the weight updates at pre and pos-synaptic neurons in order to allow network to put whatever information it needs (e.g. delayed information or gradient like) for each operation.

## 6 RESULTS

We would like to investigate the following questions. Does the system use the hidden layer? Is the system able to train with a large hidden layer size (is it stable) and does increasing size improve performance? Can the system learn if the signal is separated by larger timescales? What functional forms for the activations update and weights update work well? How does meta-gradients compare to evolution as an optimizer for update rules? How does memorization performance compare to LSTM for different network sizes? Are we able to train end to end for the language modelling, how long sequences can we train on and what is the performance? And finally, does the evolved learning algorithm transfer to sequences with different statistics?

## 6.1 SCALING WITH HIDDEN SIZE AND SEQUENCE LENGTH

We start by investigating scaling of performance with hidden layer size on the memory task for the update rules defined in Algorithm 1 with the activations update as defined in (3) and weight update in (4) and trained with natural evolution strategies. Figure 2 Left shows the performance of the system for different sequence lengths and different hidden sizes. From these curves we see: 1. The update rules are able to perform at large hidden sizes - including up to 1000 units that we studied here; 2. The system performance improves with larger hidden size; 3. When the hidden size is greater or equal to about a third of the sequence length, the network is able to essentially perfectly memorize the sequence even for sequences of 1000 steps. 4. As the number of units decreases further, the performance degrades smoothly, and the network is able to form a compressed representation of the sequence.

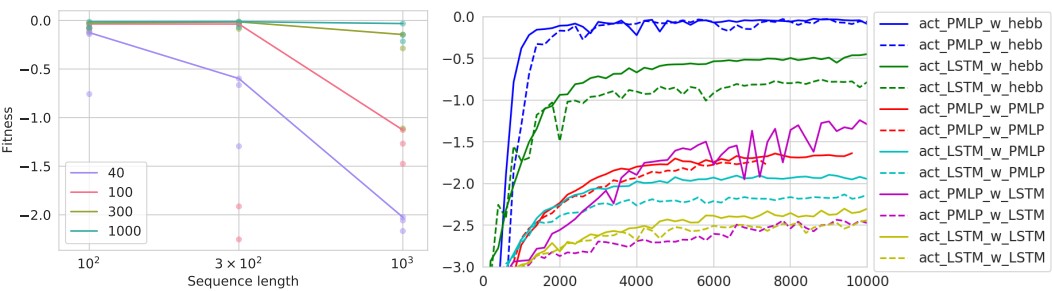

Figure 2: **Left**: Performance of the PMLP network trained with NES on the memorisation task for different sequence lengths and number of hidden units. The lines show the best performing seeds for each number of hidden units, with dots indicating individual seeds' performances. **Right**: Comparison of different functional forms for activation and weights updates on memorization of sequence of length 100. The x-axis is the evolution update number. We plot the best performing (solid) and fourth best (dashed) seed for each parameter combination. PMLP: One hidden layer neural network with $\sigma \tanh$ nonlinearity. LSTM: standard LSTM. Hebb as described in text (4). Note that the non-Hebb weight updates are significantly more expensive computationally.

## 6.2 COMPARING FUNCTIONAL FORMS

In this subsection we study the performance of different functional forms for the activations and weights updates. For the activations update we compare: Network with $\sigma \times \tanh$ non-linearity (PMLP) (3), standard MLP (different non-linearities) and LSTM. Note that this is an LSTM controlling the updates, but the base network is not an LSTM. For the weights update we compare: Hebbian update (4), MLP with $\sigma \times \tanh$ non-linearity and an LSTM. We find that the combination of PMLP with the Hebbian update works significantly better than the other combinations. Plotted in Figure 2 Right is a comparison for a simple problem of memorizing sequences of length 100 with 160 units as harder problems the other combinations fail or are too slow). As can be seen, the chosen combination performs significantly better. Furthermore, using a neural network for the weight update (MLP or LSTM) instead of the Hebbian update significantly increases computation since this network is applied at every single weight ($\sim N^2$ times where $N$ is the number of neurons).

We experimented with different ways of using the MLP and LSTM for the weight updates. First we have a choice of concatenating the input and output activations (we use 3rd and 4th component as discussed above) versus inputting their outer product (the same product used in the Hebbian update). We sweep over combinations of concatenating, outer product and both. In the PMLP case, we also investigated using the output of the network directly for the new weight, as opposed to adding the output to the old weight to get the new weight.

Taking the best combination (PMLP for activations and Hebbian for weights), we replace PMLP by a regular MLP. All runs fail with relu non-linearity. Tanh non-linearity solves the 100 steps memory task, but does not succeed on task of length 300 completely, and a larger proportion of runs fail.

## 6.3 Optimizer

In the previous experiments we found the meta-parameters $\theta$ of the PMLP network by NES. In this section we compare this to a meta-gradient optimizer, for which we simply back-propagate the objective through the entire sequence to calculate gradients of $\theta$ and use an Adam optimizer to optimize for $\theta$. Figure 3 Left shows the results. We see that meta-gradients struggles with this problem and is only able to solve relatively short sequences.

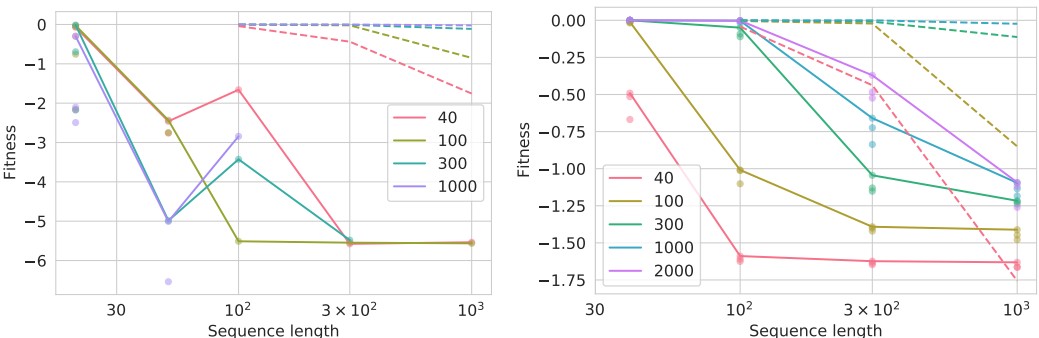

Figure 3: **Left**: Performance of meta-gradient optimisation of the PMLP network on the memorisation task for different sequence lengths. The lines show the best performing seeds for each number of hidden units, with dots indicating individual seeds' performances. The dashed lines reproduce the equivalent best performing seeds under NES optimisation shown in Fig 2. We ran all combinations of sequence length and hidden units that were possible given device memory constraints. **Right**: Performance of an LSTM network trained by meta-gradients on the memorisation task. The lines show the best performing seeds for each number of hidden units, with dots indicating individual seeds' performances. In order to facilitate comparison with a PMLP network with the same number of hidden units (but a very different architecture), the dashed lines reproduce the best performing seeds from Fig 2.

## 6.4 Comparison to an LSTM network

Next, we evaluate the performance of a standard LSTM network on the memorisation task (Fig 3 Right). The LSTM is trained in the standard way on the task: weights are kept constant within the sequence and updated across sequences - they are trained to make activations memorize the sequence. Therefore they correspond to hyper-weights of the update rules network. The LSTM has much more of these than the update rules network and over the course of training they will learn a language model. The activations of the LSTM correspond to the weights and activations of the update rules network (in that they change during the episode and are reset between episodes).

We find that meta-gradients are more effective than NES for training the LSTM network, likely because it has many more parameters than the small number of hyper-parameters $\theta$ of the PMLP update rules. We find that while LSTM can handle this task to an extent, it is unable to achieve a good performance of sequence of length 1000, while the update rules network can solve this task near perfectly.

## 6.5 Online language modelling

In this section we study how well can the system evolve, end to end, a learning algorithm for language modelling. In Figure 4 Left we show the performance for different text lengths and network sizes. As expected we see that the performance increases with sequence length as there are more dependencies that can be captured and more time to learn. In addition, importantly, we also observe that the performance increases with network size for long enough sequence lengths (2k, 10k 100k) where there are enough dependencies to be captured to benefit from larger hidden size, showing that the update rules are able to utilize the network size (10k steps, 1k units hasn't converged yet).

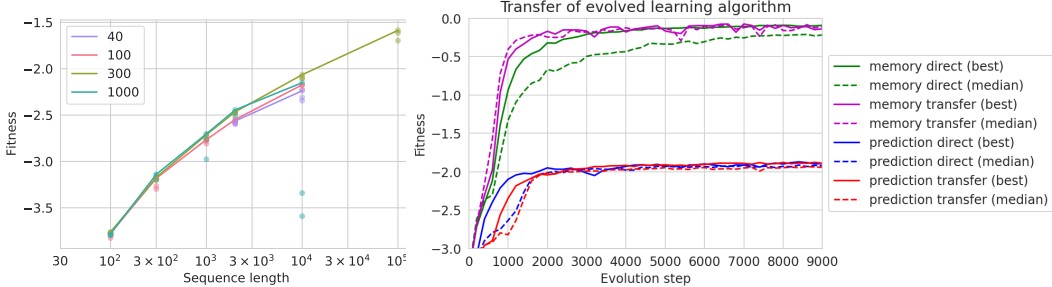

Figure 4: **Left:** End-to-end language modelling. As expected the prediction performance increases as the length increases - there are more dependencies that can be captured in longer sequences. We also observe that for sequences of 2k and 10k, larger hidden sizes generally improve the performance (the 1k units 10k length has not converged, see Figure 7 in the appendix). This shows that the algorithm is able to utilize network capacity for language modelling. For shorter sequences where there is not much structure to be captured, large number of hidden units is likely not needed. **Right:** Transfer experiment. Comparison of performance on "Alice's Abenteuer im Wunderland" in German when the algorithm was evolved directly on this text (direct) and when evolved on "Wizard of Oz" in English (transfer) on both memory and language modeling (prediction) task. We see that there is a very little difference between the original and transfer, showing that the evolved (learning) algorithm transfers well.

## 6.6 Transfer

Here we test whether evolved update rules capture a more general learning algorithm that can be applied on sequences of different statistics. For this purpose we evolve the model on the English text of "Wizard of Oz" and then test it on German text "Alice's Abenteuer im Wunderland". We compare the resulting performance to that obtained by evolving on the latter text directly. Figure 4 Right, shows the result for memorization (300 sequence length, 200 hidden units) and language modeling tasks (10000 sequence length, 300 hidden units). We see very little difference between the transfer and direct training, showing that the algorithm transfers well. (Note: In this version we converted to lower case 27 characters, so the numerical values are somewhat different)

## 6.7 Dependence on hyper-parameters

We have also studied the dependence of performance on the basic parameters of the network, the results are in the Appendix.

## 7 Discussion and Conclusions

We have succeeded at evolving online neural update rules, end to end on short-scale language modelling and somewhat difficult length sequence memorization. The main features were the following: 1. Using natural evolution strategies instead of back-propagation; 2. Using four different components of a neuron's activation vector for different purposes; 3. Using a simple Hebbian update for weights obtained from components of the activations tensor; 4. Using $\sigma \times \tanh$ non-linearity for neural update; 5. Using matrix multiplication by both weights and weights transpose between layers; 6. Using direct normalizing non-linearity at the output instead of softmax.

There are several possible future directions. One is simply scaling, as most of the experiments were run on a single GPU (per curve), while evolution is highly scalable. Achieving direct evolution on Hutter prize could be considered a next grand challenge for this approach. Another direction is to evolve functional forms as well. Finally, as discussed in the introduction, this work could have applications for building specialized neural hardware, modelling brain circuits and evolving artificial life.

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

# A    APPENDIX

## A.1    DEPENDENCE ON HYPER-PARAMETERS

We look how the performance depend on size of the neuron's state and the size of the hidden layer in neuron's activation's update. We choose a problem that is challenging but not too challenging so that large number of sweeps can be run - memorization of sequence length of 300 with network of 100 neurons. The performance is shown in Figure 6. We see that for 100 networks in the population, the best performance happens at around 10 state size and 14 hidden size while for larger population, where NES signal is smoother, the network can optimize better larger hidden sizes (of neuron's update, not the overall network size).

## A.2    LANGUAGE MODEL TRAINING CURVES

Figure 7 shows the training curves for sequence lengths 2k and 10k.

## A.3    NATURAL EVOLUTION STRATEGIES

Our implementation of separable natural evolution strategies (SNES) is summarised in Algorithm 1 (lightly modified from Wierstra et al. (2014)). At each iteration, we generate a population of $n_{\text{samples}}$ perturbed samples of the update rules parameters $\theta$, based on the current SNES search distribution. In our case, this is parameterised as a set of separable normal distributions with means $\boldsymbol{\mu}$ and log-standard deviations $\log \boldsymbol{\sigma}$. For each sample, we measure the loss $L(\boldsymbol{z}_k)$. We use the same rank-based fitness function as in Wierstra et al. (2014) Section 3.1 to convert this into a utility $u_k$. The hyper-parameters we used are summarised in Table 1.

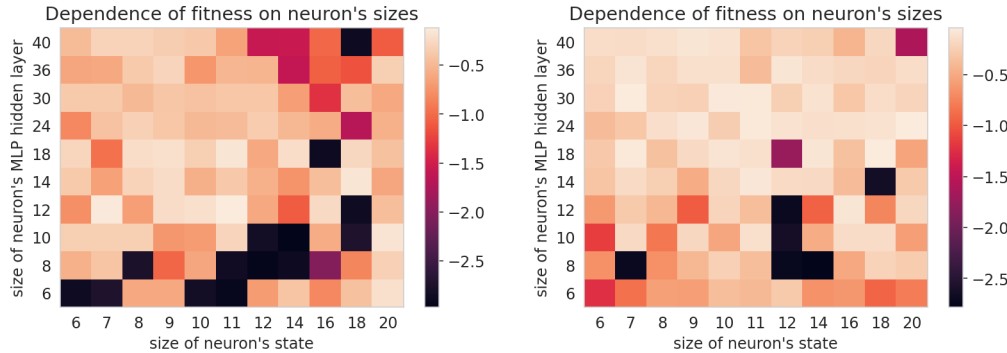

Figure 5: Dependence of fitness on neuron's hidden size and neuron's (activation's) update hidden layer size. Left vs Right: 100 vs 300 networks in NES optimizer. Only some experiments succeed at finding the solution, those that show good performance. However, more succeed with 300 networks. An improvement to finding rules algorithm would also include evolution over parallel running networks, selecting networks (update rules) that find good solutions.

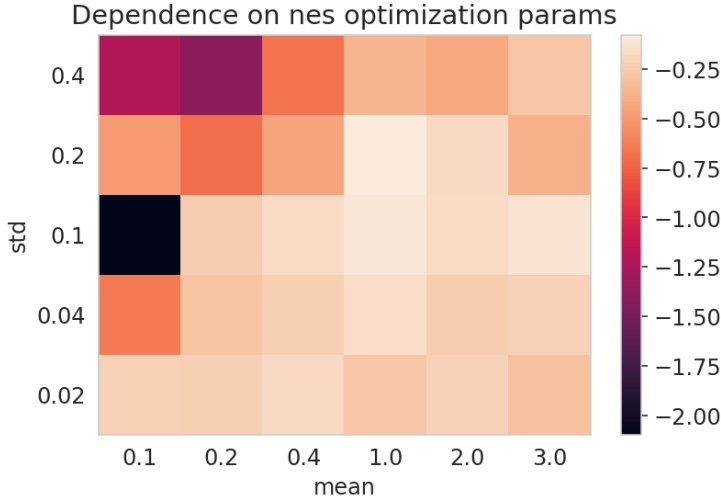

Figure 6: Dependence on NES mean and std parameters. In the experiment, the best performing update rules were run on memorization of sequence length 300, with 100 hidden units. The point in time was taken early enough where difference between the performance can be seen well. We see that std=0.1 and mean=1.0 work most robustly.

Table 1: Separable NES (SNES) hyper-parameters.

| Hyper-parameter | Value |
|---|---|
| $\eta_{\boldsymbol{\mu}}$ | 1.0 |
| $\eta_{\boldsymbol{\sigma}}$ | 0.1 |
| $\boldsymbol{\mu}_0$ | $0.0 \cdot \mathbb{I}$ |
| $\boldsymbol{\sigma}_0$ | $0.1 \cdot \mathbb{I}$ |
| $n_{\text{samples}}$ | sweep over $\{128, 256\}$ |

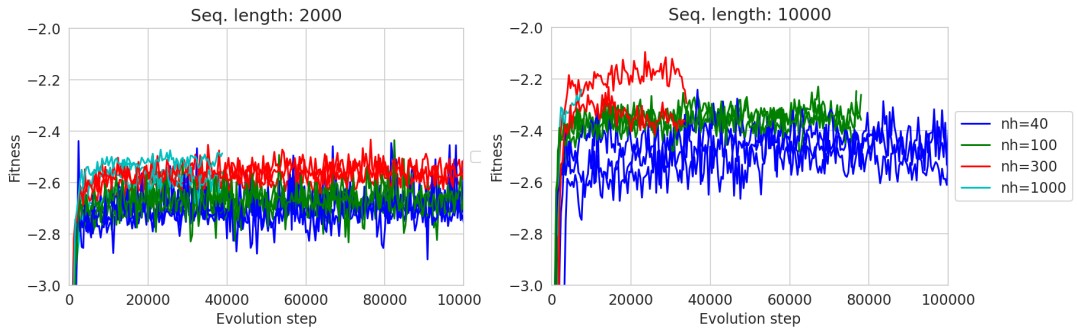

Figure 7: Language model training curves

---

**Algorithm 1** Separable NES (SNES) update step

---

**Require:** Input $\boldsymbol{\mu}_t$, $\log \boldsymbol{\sigma}_t$, $L$
    **for** $k = 1 \dots n_{\text{samples}}$ **do**
        Draw sample $\boldsymbol{s}_k \sim \mathcal{N}(0, \mathbb{I})$
        $\boldsymbol{z}_k \leftarrow \boldsymbol{\mu}_t + \boldsymbol{\sigma}_t \boldsymbol{s}_k$
        Evaluate the loss $L(\boldsymbol{z}_k)$
    **end for**
    sort $\{(\boldsymbol{s}_k, \boldsymbol{z}_k)\}$ with respect to $L(\boldsymbol{z}_k)$ and compute utilities $u_k$

    compute natural gradients   $\begin{aligned} \nabla_{\boldsymbol{\mu}} J &\leftarrow \sum_{k=1}^{n_{\text{samples}}} u_k \cdot \boldsymbol{s}_k \\ \nabla_{\boldsymbol{\sigma}} J &\leftarrow \sum_{k=1}^{n_{\text{samples}}} u_k \cdot (\boldsymbol{s}_k^2 - 1) \end{aligned}$

    update parameters   $\begin{aligned} \boldsymbol{\mu}_{t+1} &\leftarrow \boldsymbol{\mu}_t + \eta_{\boldsymbol{\mu}} \cdot \boldsymbol{\sigma} \cdot \nabla_{\boldsymbol{\mu}} J \\ \log \boldsymbol{\sigma}_{t+1} &\leftarrow \log \boldsymbol{\sigma}_t + \eta_{\boldsymbol{\sigma}}/2 \cdot \nabla_{\boldsymbol{\sigma}} J \end{aligned}$

---

