# OpenReview forum: "Evolving Neural Update Rules for Sequence Learning"
_ICLR.cc/2022/Conference — ICLR 2022 Submitted_

### Official Review · Reviewer_2qrk · 2021-11-02

**Correctness:** 3
**Technical Novelty And Significance:** 3
**Empirical Novelty And Significance:** 3
**Recommendation:** 5
**Confidence:** 3

**Main Review:**

My first and primary concern with this paper is that most of the experimental results are obtained on the memorization task. The results seem to indicate that the architecture and the evolved meta-parameter configuration are adequate for long working memory problems. It is very hard for me to judge how impressive this feat is. Despite its great focus on this task, little motivation and discussion is provided to understand its importance. Moreover, training details are missing. How is $\theta$ learned in this case? Are sequences presented twice during a training episode? The section on language modelling experiments is then disproportionally shorter. How does performance compare here against e.g. standard LSTMs?

My second concern is related to how the paper itself is presented. In general I enjoyed reading the paper but I think its clarity could be strongly improved:
- The paper often mixes the presentation of the model and training algorithm with remarks about results that will later appear. I found this very distracting, in particular in section 5.
- I missed a complete and concise set of equations that fully (and "quickly") defined the model. Some things are never properly defined. As an example, the delay $\tau$ is mentioned in the caption of Figure 1 and in the diagram but it is never explicitly defined.
- By far not enough detail is given when presenting variations of the model, for example involving LSTM weight or activation updates. The appendix could be used for these.
- Fitness (the y-axis of the plots) is never defined, I assume it is simply the log-likelihood?
- Details on how hyperparameters were chosen and which ranges were considered should be given for all variants (including e.g. standard LSTM models). Since it is difficult to optimize RNNs with gradient-based methods, how were these choices informed, in particular for standard models?
- While a few occurrences is understandable, there are currently too many references to data that is not shown ("we also experimented with ...").

Related to issues above, there a few design choices that are not sufficiently well motivated or compared against controls/alternatives. To give examples:
- How important is it to use the weight transpose in backward connections? It's not clear a priori that "emulating gradients" is needed or achieved by NES. And weight transposes are clearly biologically implausible.
- Since so many variations of the model can be tried, why did the authors choose to study softmax vs. norm(relu(.))? Why is this question especially interesting?

I liked the use of multi-dimensional activations that are then locally combined, as well as the use of different activation channels in the weight update. In fact, some additional biological motivation could be given here, and I think it would greatly strengthen the paper. For example, many neurons (including the ubiquituous pyramidal cell) are best modelled as multicompartmental units and synaptic plasticity does not only depend on the frequency of pre- and post-synaptic firing.

The text is slightly inaccurate at times:
- "[the total] log-likelihood" -> "negative log-likelihood"
- "[The first algorithm is] (meta)-gradient" -> "(meta)-gradient descent", or "(meta)-gradient-based optimization"?

If possible the paper should be proofread, as it contains some English mistakes.

**Summary Of The Paper:**

The authors design and apply recurrent neural network models with "fast weights" to sequence compression problems. The "slow" parameters that determine how the "fast" weights change are determined by gradient descent or evolution strategies. Experimentally, the latter method appears to perform better for the architectures designed by the authors. The paper focuses on a long-sequence working memory problem and additionally presents some results on language modeling.

**Summary Of The Review:**

Approching sequence learning problems with fast-weight architectures is interesting, but the experimental results are not yet convincing enough and the clarity of presentation should be improved.

---

### Official Review · Reviewer_92Z4 · 2021-11-02

**Correctness:** 2
**Technical Novelty And Significance:** 3
**Empirical Novelty And Significance:** 2
**Recommendation:** 3
**Confidence:** 3

**Main Review:**

While the premise of the paper is interesting, the stage the paper is in, unfortunately, seems not ready for publication. After a major rework (notation and clarity of writing) and additional experiments, it might be acceptable.


# Originality

While the paper combines existing methods (LSTM, MLP, self-attention PMLP, Hebb rule, etc.), the combination seems novel to me.

## Major
1. Given the topic and purpose of the paper, I would find a citation of “Learning to Learn Using Gradient Descent“ (Hochreiter, 2001) necessary. Especially w.r.t. section 6.4.
2. A citation of “Learning to learn by gradient descent by gradient descent” (Andrychowicz, 2016) would be appropriate w.r.t. section 1.1, paragraph “End-to-end learning and meta-learning”.

## Minor
2. The claim that “with the best models being transformers currently, and LSTM recurrent networks in the not so recent past.” is unsupported and, to my knowledge, wrong. LSTM outperform transformers on sequence learning tasks that involve integrating information over time, such as rain fall-off prediction [“Rainfall–runoff modelling using Long Short-Term Memory (LSTM) networks” (Kratzert, 2018)].


# Quality

## Major
3. If I understand correctly, the experiments were only conducted on one task (next-character prediction for books), which seems insufficient to me. I advice the authors to look into either next-character prediction for biological data (DNA, amino acids) or tasks involving math, such as simple calculations that need to be performed. Both tasks allow for very long sequence lengths. With such experiments, the empirical significance could be improved drastically.
4. When considering long sequences and comparing to LSTM networks, it is very important to clarify if a forget gate is used. A forget gate will re-introduce the vanishing gradients problem and prohibit learning on longer sequences, if not initialized with a large positive bias (so that the gate is forced “open” for many updates). I have not found information on this in the paper, so I assume a forget gate was used without consideration of vanishing gradients.

## Minor
5. “Being able to evolve learning algorithms without objective might provide a path to intelligent artificial life.” - but you do have an objective for your method, no? Also, how do you define “artificial life”?
6. Section “Tasks”: “The N is of the order of ten thousand in our experiments, but in principle it can be arbitrarily long or we should ideally use the entire data-set, this is just the scale we have managed to achieve so far.” - that is a bold claim but I don’t see any justification for it?
7. “This working relies on a generally reasonable assumption that statistics of data in the future
will be similar to the current ones. In our approach we don’t make such an assumption - the system
evolves how it should process the current piece of information in order to do well in the future.” - but that still implies that the future data is similar to the current data. Otherwise how would learning to process the current data help in the future data?
8. “[...] the search process needs to be able to find relationships that span thousands even possibly millions of steps. Thus, rather than hoping to find such signal by back-propagating over such periods, we can expect evolution to work better: For example, if local update rules are modified by chance to store information, this modification can have immediate effect over long time scales.” - I disagree – a sequence model with an attention mechanism can pinpoint a signal very well. I don’t see why evolution would be expected to work better here.
9. Please consider plotting the means of your runs with a shaded area for the error bars or best/worst runs instead of only plotting the best and 4th best seed. (It might seem cherry-picked otherwise.)
10. “The LSTM has much more of these [hyper-weights] than the update rules network[…]” - Please give some ranges here so that it is clear how many more parameters the LSTM has in your case.


# Clarity

The paper seems very unclear to me. Unfortunately, I find the notation inconsistent and hard to read.

## Major
11. The mixture of Python syntax and math formulas is not very readable. I would suggest introducing variable names or suffices instead of indexing the vectors with h[:, 0], h[:, 1],…, h[:, -1] and giving the indices different meanings.
12. Please define all abbreviations, even if they are common (MLP, LSTM).
13. Please define all symbols. (sigma, uppercase W).

## Minor
14. “For a given connected pair of layers, we do two matrix multiplications (denoted by ·): w^{lk} (that “goes” from layer k to layer l) multiplies the 0-th component h^k[:, 0] of tensor hk and the transpose of w^{kl} (that ”goes” from layer l to layer k) multiplies the 1-st component h^k[:, 1] of hk .”  - I assume you mean that w^{lk} is multiplied by h^k[:, 0]? Or am I missing something here?
15. Eq. (3) is self-attention or gating – please mention this when you introduce it, it would make understanding it much easier in my opinion.
16. Are the matrices M are trainable? They are not really described.
17. Eq. (3) and (4): I would recommend to also show the output of the equation, e.g. w_new = WU() = … .
18. I would suggest to annotate the matrices and vectors with their sizes (e.g. $w\in R ^{n \times m}$).
19. In the abstract I would suggest to write “recurrent neural network” instead of just “recurrent network” for clarity.
20. n^l and n_l are the same because of a typo, I assume?
21. Sometimes you write “sigma tanh” and sometimes “sigma x tanh”, is this on purpose?
22. “While we experimented with various forms of the update, […]” - I would suggest to list the updates you have considered in your experiments here for clarity.
23. “(10k steps, 1k units hasn’t converged yet).” - Does this mean it is still running for the rebuttal or just not finished and final?


# Significance

I cannot judge the significance of this work, mainly because of missing clarity on points 3. and 4. in “Quality”.

**Summary Of The Paper:**

This paper aims at using evolution strategies instead of back-propagation for evolving the parameters of weight and activation updates for an online-learning sequence model, specifically a next-character prediction model. Each neuron activation is represented by a vector, where each vector element has a different purpose, such as matrix multiplication for forward and multiplication with a transposed matrix for recurrent connections. LSTM, MLP, MLP with self attention/gating (PMLP), and Hebb rule were considered for activation and weight updates. Experiments were conducted on a next-character prediction task on long sequences.

**Summary Of The Review:**

While the premise of the paper is interesting, I can not recommend it for publication. The paper is very hard to read and suffers from missing clarity. The experimental evaluation is insufficient.
I would suggest to the authors that they conduct a major rework of the paper (notation and clarity of writing), add additional experiments, and submit to a later conference.

---

### Official Review · Reviewer_3DR8 · 2021-11-03

**Correctness:** 3
**Technical Novelty And Significance:** 2
**Empirical Novelty And Significance:** 2
**Recommendation:** 3
**Confidence:** 3

**Main Review:**

The paper deals with an important topic: how can we train neural update rules that are more efficient than general update rules based on approaches such as gradient descent. However, currently, the contributions of the paper are not made clear enough and some crucial comparisons are missing.

How does the approach compare to related work on evolving Hebbian learning rules with an evolutionary strategy (e.g. Najarro&Risi 2020, "Meta-Learning through Hebbian Plasticity in Random Networks")? This approach seems very related to the described work. Additionally, other relevant work that should be compared against includes the mentioned work by (Kirsch & Schmidhuber, 2020) or the more complex model of Bertens and Lee "Network of evolvable neural units can learn synaptic learning rules and spiking dynamics”.

The introduction describes the approach as a HyperNetwork, in which a trained function is responsible for the weight updates (also mentioning related work such as HyperNetworks/CPPN), but then only later in the paper is the actual Hebbian approach described. It would be good to see an actual comparison to a HyperNetwork style approach (or rewriting the paper to focus on a local-learning rule approach), even though it might be computational more expensive.

Additional comments:
- “In (Stanley & Miikkulainen, 2002) they parameterize the weights by compo- sitional pattern producing networks (CPPN)” -> CPPN were not invented at this point. They evolved the weights and architecture of networks with NEAT”

-"The N is of the order of ten thousand in our experiments, but in principle it can be arbitrarily long or we should ideally use the entire data-set, this is just the scale we have managed to achieve so far” -> if it works in principle, what are the challenges in scaling to longer sequences?

-"The standard way to compare models is on their likelihood on a withheld test set.” -> This comparison should also be included

**Summary Of The Paper:**

This paper presents an approach that trains local Hebbian learning rules to allow a neural network to perform reasonably well in two types of problems: sequence memorization and prediction. Two approaches to train these models are compared, which are based on meta-gradients and evolutionary strategies. The evolved model is able to perform sequence predictions of length 1000.

**Summary Of The Review:**

The paper presents an approach based on local learning rules to deal with two tasks: sequence memorization and prediction. The results are promising but crucial comparisons to related approaches are missing, making it difficult to judge the particular contributions of this paper.

---

### Decision · Program_Chairs · 2022-01-20

**Decision:**

Reject

**Comment:**

The paper addresses an interesting problem, namely how to evolve effective weight and activation update rules for online learning of a recurrent network. The work focuses on two specific tasks: character sequence memorisation and prediction. Two approaches based on meta-gradients and evolutionary strategies are explored. Unfortunately the paper is missing some important related works. Moreover, presentation needs to be improved, as well as experimental assessment should be expanded both in terms of tasks and in terms of comparable models presented in the literature.